# Efficacy of Wex-cide 128 disinfectant against multiple prion strains

Chase Baune[1], Bradley R. Groveman[1], Andrew G. Hughson[1], Tina Thomas[2], Barry Twardoski[3], Suzette Priola[1], Bruce Chesebro[1], Brent Race[1]*

1 The Laboratory of Neurological Infections and Immunity, Rocky Mountain Laboratories, National Institute of Allergy and Infectious Diseases, National Institutes of Health, Hamilton, Montana, United States of America, 2 Rocky Mountain Veterinary Branch, Rocky Mountain Laboratories, National Institute of Allergy and Infectious Diseases, National Institutes of Health, Hamilton, Montana, United States of America, 3 Office of Operations Management, Rocky Mountain Laboratories, National Institute of Allergy and Infectious Diseases, National Institutes of Health, Hamilton, Montana, United States of America

* raceb@niaid.nih.gov

**Data Availability Statement:** All relevant data are within the manuscript and its Supporting Information files.

**Funding:** This research was supported by the Intramural Research Program of the NIH, National

## Abstract

Prion diseases are transmissible, fatal neurologic diseases that include Creutzfeldt-Jakob Disease (CJD) in humans, chronic wasting disease (CWD) in cervids, bovine spongiform encephalopathy (BSE) in cattle and scrapie in sheep. Prions are extremely difficult to inactivate and established methods to reduce prion infectivity are often dangerous, caustic, expensive, or impractical. Identifying viable and safe methods for treating prion contaminated materials is important for hospitals, research facilities, biologists, hunters, and meat-processors. For three decades, some prion researchers have used a phenolic product called Environ LpH (eLpH) to inactivate prions. ELpH has been discontinued, but a similar product, Wex-cide 128, containing the similar phenolic chemicals as eLpH is now available. In the current study, we directly compared the anti-prion efficacy of eLpH and Wex-cide 128 against prions from four different species (hamster 263K, cervid CWD, mouse 22L and human CJD). Decontamination was performed on either prion infected brain homogenates or prion contaminated steel wires and mouse bioassay was used to quantify the remaining prion infectivity. Our data show that both eLpH and Wex-cide 128 removed 4.0–5.5 logs of prion infectivity from 22L, CWD and 263K prion homogenates, but only about 1.25–1.50 logs of prion infectivity from human sporadic CJD. Wex-cide 128 is a viable substitute for inactivation of most prions from most species, but the resistance of CJD to phenolic inactivation is a concern and emphasizes the fact that inactivation methods should be confirmed for each target prion strain.

## Introduction

Prion diseases, also known as transmissible spongiform encephalopathies (TSE), are unique infectious diseases that occur following the repeated conversion of normal host derived cellular prion protein (PrPC) into a mis-folded, protease-resistant, infectious, disease associated conformation (PrPSc) [1]. Unfortunately, infectious prions are inherently difficult to inactivate

Institute of Allergy and Infectious Diseases. The funders had no role in study design, data collection and analysis, decision to publish, or preparation of the manuscript.

**Competing interests:** The authors declare that they have no competing interests.

and have posed a biosafety challenge for research laboratories, medical facilities, and meat processing plants for many years. Common physical and chemical methods for destruction of bacterial and viral pathogens are not effective in eliminating prion infectivity. However, several chemical disinfectants, including concentrated sodium hypochlorite (bleach) sodium hydroxide (NaOH), and Environ LpH (eLpH) have been identified that do inactivate prions [2–4]. Only eLpH, bleach and NaOH are included as "Prion Inactivation Methods for Reusable Instruments and Surfaces" in the 5th and 6th editions of the Biosafety in Microbiological and Biomedical Laboratories published by the CDC and NIH. Of the three chemical inactivation options, each has its pros and cons.

Environ LpH is much less caustic to equipment and not as toxic or cumbersome to handle and discard appropriately. Importantly, eLpH has the ability to remove greater than $10^7$ $LD_{50s}$ of prion infectivity from 263K scrapie infected hamster brain homogenate [2]. The mode of action for eLpH against prion inactivation was never identified, but research on several other phenolic products in the LpH series showed poor anti-prion activity [4]. Unfortunately, production of eLpH has been discontinued. However, a similar phenolic product, which includes two of the same phenols (Ortho-benzyl-para-chlorophenol (BP) and O-phenylphenol (OPP)) present in eLpH is now available from Wexford Labs marketed as Wex-cide 128. Wex-cide 128 is an EPA registered pesticide marketed as a disinfectant, deodorizer and cleaner for healthcare, schools, and industry. We were interested in the potential of Wex-cide 128 as a prion disinfectant.

In the current study, we compared the efficacy of Wex-cide 128 to eLpH against infectious prions derived from four different species: hamster adapted 263K scrapie, white-tailed deer Chronic Wasting Disease (CWD), mouse-adapted scrapie strain 22L, and one human prion strain, MM1 sporadic Creutzfeldt-Jakob Disease (sCJD). We tested all four prion strains by decontaminating prion infected brain homogenates followed by animal bioassay to measure remaining prion infectivity. For two of the strains, 263K and sCJD, we also tested inactivation of prions bound to steel wires. Steel wires act as a surrogate for surgical instruments and have a non-porous surface similar to many coatings present in laboratories and hospitals. Our results using mouse bioassays showed that both eLpH and Wex-cide 128 were highly efficacious and suitable for inactivation of CWD, 22L and 263K prions, but much less effective against human sCJD. The discovery that sCJD was more resistant to inactivation by phenolic disinfectants is a concern and reaffirms that prion disinfectant efficacy must be verified for each target prion strain/species, as not all infectious prions are inactivated equally [5–10].

## Results

### Efficacy and shelf life of Wex-cide 128 against 263K hamster prions

Wex-cide 128 is typically used at a 1:128 dilution ($\sim$0.8%) for general disinfectant applications. However, in our studies we tested a 4% dilution of Wex-cide 128 in order to normalize the BP concentration to what is present in 2% eLpH (Table 1). Environ LpH has previously established efficacy against 263K hamster prions and is routinely used in our laboratory as a 2% solution to inactivate prions. We have included 2% eLpH in the current study as a prion inactivation control and experimental group for historical/experimental comparison. We also tested Wex-cide 128 at a ten-fold higher concentration (40%) to better understand the level of phenols necessary for anti-prion activity. Ten percent 263K-infected brain homogenates were mixed at a 1:9 ratio of brain homogenate to disinfectant for 30 minutes. After this decontamination step, the brain homogenate/disinfectant mixture was further diluted and immediately inoculated intracerebrally into tg7 mice. Additional dilution was necessary to prevent acute toxicity in recipient bioassay mice due to the residual disinfectant. As a no treatment control,

**Table 1. Chemical composition of undiluted, stock phenolic disinfectants.**

| Ingredient | Environ-LpH (% by weight) | Wex-Cide 128 (% by weight) |
|---|---|---|
| Ortho-benzyl-para-chlorophenol | 6.4 | 3.03 |
| o-Phenylphenol | 0.5 | 3.4 |
| Hexylene glycol | 4 | 10–30 |
| Isopropanol | 8 | 1–5 |
| Glycolic Acid | 12.6 | 0 |
| P-tertiary-amylphenol | 3 | 0 |

263K brain homogenate was treated with saline for 30 minutes prior to dilution and inoculation.

Our data showed 4% Wex-cide 128 and 2% eLpH both reduced 263K infectivity by over 5 logs compared to saline treatment alone (Table 2). Importantly, our data also showed no added benefit to using 40% Wex-cide over 4% against 263K prions. Interestingly, one mouse inoculated with a $10^{-3}$ dilution of eLpH treated 263K did develop prion disease at a late time (Table 2). To our knowledge, this is the first time eLpH failed to inactivate prions in brain homogenate to below detectable limits, [2, 4, 11] but the stock eLpH used for these studies was at least a decade post-manufacture and may have lost full efficacy.

To test the shelf-life of Wex-cide 128, we performed similar prion inactivation experiments to those described above using disinfectants that had been diluted and aged. Wex-cide 128 and eLpH were diluted to 4% and 2% respectively and kept on the laboratory bench for either 6 weeks or 8 months at room temperature and natural light conditions. After aging, the diluted disinfectants were used to decontaminate 263K brain homogenates and the treated homogenates were inoculated into recipient tg7 mice to detect remaining prion infectivity. Mice in this experiment were observed up to 299 days post-inoculation (dpi). During this observation time, no mice developed signs of clinical disease. Following euthanasia, brains from three mice that appeared normal at the termination of the experiment, tested positive by immunoblot.

**Table 2. Bioassay of disinfected 263K brain homogenate in tg7 mice.**

| Disinfectant (fresh) | Dilution of 263K scrapie brain homogenate inoculated after treatment[a] | | | | | | | Titer[b] | Log$_{10}$ Reduction in titer |
|---|---|---|---|---|---|---|---|---|---|
| | $10^{-3}$ | $10^{-4}$ | $10^{-5}$ | $10^{-6}$ | $10^{-7}$ | $10^{-8}$ | $10^{-9}$ | | |
| Saline | 4/4 [c], 55 | 4/4, 62 | 4/4, 66 | 4/4, 71 | 4/4, 81.5 | 1/4, 258 | 1/4, 146 | 9.5 | NA |
| 40% Wex-cide | nt | 0/5 | 0/4 | 0/4 | nt | nt | nt | ≤ 5.0 | **≥4.5** |
| 4% Wex-cide | 0/6 | 0/3 | 0/4 | 0/3 | nt | nt | nt | ≤ 4.0 | **≥5.5** |
| 2% eLpH | 1/4, 197 | 0/4 | 0/4 | nt | nt | nt | nt | 4.25 | **5.25** |
| Disinfectant (aged) | | | | | | | | | |
| 4% Wex-cide 6 weeks | 0/8, 288 | 0/6 | 0/5 | nt | nt | nt | nt | ≤ 4.0 | **≥5.5** |
| 2% eLpH 6 weeks | 0/8, 288 | 0/7 | 0/6 | nt | nt | nt | nt | ≤ 4.0 | **≥5.5** |
| 4% Wex-cide 8 months | 0/8, 299 | 0/6 | 0/7 | nt | nt | nt | nt | ≤ 4.0 | **≥5.5** |
| 2% eLpH 8 months | 0/8, 299 | 0/9 | 0/5 | nt | nt | nt | nt | ≤ 4.0 | **≥5.5** |

[a]Aliquots of 263K brain homogenates (10%) were exposed to different disinfectants or saline for 30 minutes at a 1:9 ratio. Solutions were then further diluted for bioassay in mice. Each recipient mouse received 30μl of inoculum.

[b] The calculated titer reported is the log$_{10}$LD$_{50}$/gram of tissue

[c] The numerator is the number of prion-positive mice (see Methods), and the denominator is the number of mice inoculated. For groups with positive mice the average incubation period in days-post inoculation (dpi) is provided. Tg7 mice typically do not develop 263K prion disease after 200 dpi. Mice that did not develop clinical signs of prion disease were euthanized at 288–300 dpi.

NA: not applicable, nt: not tested

These subclinical infections occurred in one mouse from each of the $10^{-3}$ Wex-cide groups and one mouse from the eLpH that had been aged 6 weeks (S1 Table). Since these mice did not meet our full criteria for scoring positive (see Methods), they have been excluded from Table 2. Using the aged disinfectants, 263K prion infectivity was again reduced by over 5 logs by both 4% Wex-cide 128 and 2% eLpH, demonstrating that both phenolic mixtures have stability to at least 8 months post-dilution (Table 2).

## Efficacy of Wex-cide 128 against stainless-steel bound 263K hamster prions

We next studied whether Wex-cide 128 could inactivate prions that were bound to steel surfaces. The ability of a disinfectant to eliminate dried prions from surfaces is an important consideration for any chemical that may be used as a prion decontaminant. As a surrogate for a stainless-steel surface, we used 3–4 mm segments of stainless-steel wire suture. Wires coated with 263K prions (see Methods) were immersed in either 4% Wex-cide 128 or 2% eLpH for either 2 minutes or 30 minutes. Two minutes was tested as a reasonable contact time for a disinfectant applied to a hard surface such a biosafety cabinet or countertop. Thirty minutes simulated an immersion situation, where instruments or tools could be placed in a container of disinfectant. Following decontamination wires were rinsed briefly with distilled water and allowed to air dry. As a positive control, and to provide an estimate of how much prion infectivity could be maximally bound to the wires, we also immersed groups of wires in serial ten-fold dilutions of 263K brain homogenate. Positive control (no disinfectant treatment) and treated wires were implanted into the brains of recipient tg7 mice as a biological indicator of prion infectivity. None of the mice implanted with Wex-cide or eLpH treated wires developed prion disease after a 300-day observation period (Table 3). Wires that were exposed to serial dilutions of 263K prions caused clinical disease in tg7 mice with incubation periods that correlated closely with the concentration of 263K used to coat the wire (Table 3). If we assume that the prion infectivity on the wire correlates to the exposure dose, we estimate that over 6 logs of infectivity was inactivated by the Wex-cide and eLpH treatments, even with as little as a 2-minute exposure time.

**Table 3. Bioassay of disinfected 263K coated wires in tg7 mice.**

| Disinfectant | Exposure Time (min) | Dilution of 263K brain homogenate used to coat wires, prior to treatment[a] | | | | | Titer[b] | Log₁₀ Reduction in titer |
| --- | --- | --- | --- | --- | --- | --- | --- | --- |
| | | $10^{-1}$ | $10^{-4}$ | $10^{-5}$ | $10^{-6}$ | $10^{-7}$ | | |
| None | NA | 5/5 [c], 65 | 4/4, 83 | 4/4, 95.5 | 5/5, 142 | 1/8, 125 | 6.6 | NA |
| 4% Wex-cide | 2 | 0/4 | nt | nt | nt | nt | < 0.5 | ≥6.1 |
| 4% Wex-cide | 30 | 0/4 | nt | nt | nt | nt | < 0.5 | ≥6.1 |
| 2% eLpH | 2 | 0/7 | nt | nt | nt | nt | < 0.5 | ≥6.1 |
| 2% eLpH | 30 | 0/6 | nt | nt | nt | nt | < 0.5 | ≥6.1 |

[a] Steel wires were exposed to 263K prion infected brain homogenates, then washed, dried, and immersed in different disinfectants for either 2 or 30 minutes. Following treatment wires were removed and allowed to dry. Each mouse was implanted intracerebrally with a single 3–4 mm wire.

[b] The calculated titer reported is the $log_{10}LD_{50}$ / wire

[c] The numerator is the number of prion-positive mice (see Methods), and the denominator is the number of mice implanted. For groups with positive mice the average incubation period in days-post inoculation (dpi) is provided. Tg7 mice typically do not develop 263K prion disease after 200 dpi. Mice in this experiment that did not develop clinical signs of prion disease were euthanized at 315 dpi.

NA: not applicable, nt: not tested

## Efficacy of Wex-cide 128 against cervid-derived CWD prions and 22L rodent-adapted mouse scrapie prions

Having demonstrated efficacy against hamster 263K prions, we then tested the ability of 4% Wex-cide 128 to remove prion infectivity from both cervid-derived CWD and rodent-adapted scrapie (strain 22L) prion infected brain homogenates. None of the mice inoculated with CWD infected brain homogenates treated with 4% Wex-cide 128 or 2% eLpH for 30 minutes developed prion disease. This indicates that at least 4.77 logs of CWD prion infectivity (Table 4) were removed with Wex-cide treatment. Decontamination of 22L-infected mouse brain homogenates did not eliminate all the prion infectivity from the brain homogenates. In mice inoculated with treated $10^{-3}$ 22L-infected mouse brain homogenates, 1/8 mice in the Wex-cide group and 8/8 mice in the eLpH group developed clinical signs of prion disease. Additionally, 2/7 mice inoculated with $10^{-4}$ eLpH treated brain homogenate also became clinical. Even with this evidence of residual prion infectivity, both treatments demonstrated efficacy with 4% Wex-cide removing over 5 logs of 22L prion infectivity and 2% eLpH removing 4 logs (Table 5).

## Efficacy of Wex-cide 128 against sCJD brain homogenates and prion coated steel wires

Decontamination of human tropic prions is an important biomedical and research biosafety concern. We therefore tested the ability of Wex-cide 128 and eLpH to inactivate sCJD prions derived from transgenic mice that expressed human prion protein with methionine at codon 129. We tested inactivation of sCJD in brain homogenates and also bound to stainless steel wires. We found that sCJD brain homogenates treated for 30 minutes with 2% eLpH or 4% Wex-cide 128 showed only a slight reduction in prion infectivity of 1.25 logs (Table 6). Using a ten-fold higher concentration of Wex-cide 128 only improved the reduction in sCJD prion infectivity by an additional 0.25 logs. Inactivation of steel wire bound sCJD prions was more effective. We achieved complete removal of prion infectivity when the sCJD coated wires were immersed for 30 minutes in either eLpH or Wex-cide 128 (Table 7). Two minutes in these same disinfectants was not as effective, as several mice with treated, implanted wires developed sCJD (Table 7). In contrast to the 263K coated wires shown in Table 3, wires coated with 10-fold serial dilutions of sCJD did not correlate with increasing incubation periods as the wires were exposed to less sCJD (Table 7). Because of this non-linear response, and failure to

**Table 4. Bioassay of disinfected CWD brain homogenate in tg33 mice.**

| Disinfectant | Dilution of CWD brain homogenate inoculated after treatment[a] | | | | | Titer[b] | Log$_{10}$ Reduction in titer |
|---|---|---|---|---|---|---|---|
| | $10^{-3}$ | $10^{-4}$ | $10^{-5}$ | $10^{-6}$ | $10^{-7}$ | | |
| Saline | 3/3 [c], 297 | nt | 4/4, 440 | 4/4, 379 | 3/4, 533 | $\geq 8.77$ | NA |
| 4% Wex-cide | 0/5 | 0/4 | 0/4 | nt | nt | $\leq 4.0$ | $\geq \mathbf{4.77}$ |
| 2% eLpH | 0/5 | 0/4 | 0/4 | nt | nt | $\leq 4.0$ | $\geq \mathbf{4.77}$ |

[a] Aliquots of CWD brain homogenates (10%) were exposed to different disinfectants or saline for 30 minutes at a 1:9 ratio. Solutions were then further diluted for bioassay in mice. Each recipient mouse received 30μl of inoculum.

[b] The calculated titer reported is the $\log_{10}LD_{50}$ / gram of tissue. Calculation of the titer for the saline treated group assumed no positive mice occurred at a $10^{-8}$ dilution since we did not reach an end-point.

[c] The numerator is the number of prion-positive mice (see Methods), and the denominator is the number of mice inoculated. For groups with positive mice the average incubation period in days-post inoculation (dpi) is provided. Tg33 mice typically do not develop CWD prion disease after 600 dpi. Mice that did not develop clinical signs of prion disease were euthanized at 650 dpi.

NA: not applicable, nt: not tested

**Table 5. Bioassay of disinfected 22L brain homogenate in tga20 mice.**

| Disinfectant | Dilution of 22L brain homogenate inoculated after treatment[a] | | | | | | | Titer[b] | Log₁₀ Reduction in titer |
| | $10^{-3}$ | $10^{-4}$ | $10^{-5}$ | $10^{-6}$ | $10^{-7}$ | $10^{-8}$ | $10^{-9}$ | | |
|---|---|---|---|---|---|---|---|---|---|
| Saline | 4/4 [c], 94 | 4/4, 104 | 4/4, 112 | 4/4, 126 | 4/4, 156 | 1/4, 149 | 0/4 | 9.25 | NA |
| 4% Wex-cide | 1/8, 228 | 0/8 | nt | nt | nt | nt | nt | 4.15 | **5.1** |
| 2% eLpH | 8/8, 158 | 2/7, 174 | nt | nt | nt | nt | nt | 5.3 | **4.0** |

[a]Aliquots of 22L-infected brain homogenates (10%) were exposed to different disinfectants or saline for 30 minutes at a 1:9 ratio. Solutions were then further diluted for bioassay in mice. Each recipient mouse received 30μl of inoculum.

[b] The calculated titer reported is the $\log_{10}LD_{50}$ / gram of tissue

[c] The numerator is the number of prion-positive mice (see Methods), and the denominator is the number of mice inoculated. For groups with positive mice the average incubation period in days-post inoculation (dpi) is provided. Tga20 mice typically do not develop 22L prion disease after 200 dpi. Mice that did not develop clinical signs of prion disease were euthanized at 289 dpi.

NA = not applicable, nt = not tested

reach an end-point in our no treatment control we did not attempt to extrapolate a decrease in titer for this experiment.

## Discussion

For decades prion researchers have used eLpH as an effective alternative to bleach or NaOH to chemically inactivate prions. After eLpH became unavailable we sought to find a suitable acidic phenol that had comparable or increased prion inactivation ability. Wex-cide 128 was a clear candidate to screen as it contained the same two phenols that are the main components in eLpH (Table 1).

In our experiments, a 30-minute treatment using 4% Wex-cide 128 reduced prion infectivity from brain homogenates by at least $10^4$ infectious units for 263K, 22L and CWD (Table 8). Many of these reductions are likely to be underestimations as our lower limit of detection in the assay is dictated by acute toxicity from the disinfectant and prevents us from testing more concentrated samples. Wex-cide 128 treatment also removed at least $10^6$ infectious units from 263K coated steel wires (Table 3). Compared to eLpH, Wex-cide 128 was equal to or superior to eLpH in reduction of infectious prions derived from four different species (Table 8). Wex-cide 128 also demonstrated stable shelf life of at least 8 months following dilution to a 4%

**Table 6. Bioassay of disinfected sCJD brain homogenate in tg66 mice.**

| Disinfectant | Dilution of sCJD brain homogenate inoculated after treatment[a] | | | | | Titer[b] | Log₁₀ Reduction in titer |
| | $10^{-3}$ | $10^{-4}$ | $10^{-5}$ | $10^{-6}$ | $10^{-7}$ | | |
|---|---|---|---|---|---|---|---|
| Saline | 4/4 [c], 204 | 4/4, 296 | 3/4, 416 | 0/4 | 0/4 | 6.77 | NA |
| 40% Wex-cide | nt | 1/5, 434 | 0/4 | 0/4 | nt | 5.22 | **1.5** |
| 4% Wex-cide | 4/4, 309 | 2/4, 372 | 0/4 | 0/5 | nt | 5.52 | **1.25** |
| 2% eLpH | 3/4, 453 | 3/4, 481 | 0/4 | nt | nt | 5.52 | **1.25** |

[a]Aliquots of sCJD prion infected brain homogenates (10%) were exposed to different disinfectants or saline for 30 minutes at a 1:9 ratio. Solutions were then further diluted for bioassay in mice. Each recipient mouse received 30 μl of inoculum.

[b] The calculated titer reported is the $\log_{10}LD_{50}$ / gram of tissue

[c] The numerator is the number of prion-positive mice (see Methods), and the denominator is the number of mice inoculated. For groups with positive mice the average incubation period in days-post inoculation (dpi) is provided. Tg66 mice typically do not develop sCJD after 550 dpi. Mice that did not develop clinical signs of prion disease were euthanized at 650 dpi.

NA = not applicable, nt = not tested

**Table 7. Bioassay of disinfected sCJD coated wires in tg66 mice.**

| Disinfectant | Exposure Time (min) | Dilution of sCJD brain homogenate used to coat wires, prior to treatment[a] | | | |
|---|---|---|---|---|---|
| | | $10^{-1}$ | $10^{-3}$ | $10^{-4}$ | $10^{-5}$ |
| None | NA | 4/4 [c], 319 | 2/3, 311 | 4/4, 348 | 4/4, 360 |
| 4% Wex-cide | 2 | 1/6, 496 | nt | nt | nt |
| 4% Wex-cide | 30 | 0/6 | nt | nt | nt |
| 2% eLpH | 2 | 5/6, 391 | nt | nt | nt |
| 2% eLpH | 30 | 0/6 | nt | nt | nt |
| Water | 30 | 6/6, 303 | nt | nt | nt |

[a] Steel wires were exposed to sCJD prion infected brain homogenates, then washed, dried, and immersed in different disinfectants for either 2 or 30 minutes. Following treatment wires were removed and allowed to dry. Each mouse was implanted intracerebrally with a single 3–4 mm wire.

[b] The calculated titer reported is the $\log_{10}LD_{50}$ / wire

[c] The numerator is the number of prion-positive mice (see Methods), and the denominator is the number of mice inoculated. For groups with positive mice the average incubation period in days-post inoculation (dpi) is provided. Tg66 mice typically do not develop sCJD after 550 dpi. Mice that did not develop clinical signs of prion disease were euthanized at 650 dpi.

NA = not applicable, nt = not tested

working concentration (Table 2). Collectively, we believe our data show that 4% Wex-cide 128 is a valid replacement for 2% eLpH for use against 263K, 22L and CWD prions. Beyond the research laboratory, Wex-cide 128 provides a less dangerous and less corrosive option compared to concentrated bleach, and may be useful for wildlife biologists, meat processors, and hunters handling CWD [12].

The utility of wires as a surrogate for surfaces and instruments have been used with success by several other groups testing prion inactivation [5, 8, 13–17]. Our studies using 263K coated steel wires had a clear advantage in sensitivity over the homogenate bioassays (Tables 2 and 3). Following prion coating and subsequent decontamination, the steel wires can be dried, effectively eliminating residual decontaminate solution. This feature allows higher sensitivity, as decontaminated brain homogenates must be diluted prior to inoculation in mice to avoid acute toxicity. For our decontaminated wire groups, we tested only wires that had been immersed in 10% brain homogenates, rather than a full titration. Absence of infectivity at this single dilution was therefore based on the outcome of only 4–7 wire implanted mice per condition. If a low level of infectivity did still exist on a small subset of decontaminated wires, testing a larger group size would provide more confidence that all infectivity was eliminated.

In the current study we found that 263K prions and sCJD prions appear to have differing affinities for binding steel. Bioassay data from implanted wires coated with 10-fold decreasing concentrations of 263K dilutions showed that wires exposed to fewer 263K prions bound less infectious 263K prions based on incubation periods in mice (Table 3). This data was also similar to previous studies using 263K or vCJD coated wires [9, 11]. The same trend was not observed in our wire experiments with sCJD, where decreasing concentrations of sCJD did

**Table 8. Summary of 4% Wex-cide 128 and 2% eLpH inactivation of prions in brain homogenates.**

| Disinfectant | Reduction in prion titer ($\log_{10}$) for each strain of prion | | | |
|---|---|---|---|---|
| | 263K (9.5)[1] | CWD (8.77)[1] | 22L (9.25)[1] | sCJD (6.77)[1] |
| 4% Wex-cide | $\geq 5.5$ | $\geq 4.77$ | 5.1 | 1.5 |
| 2% eLpH | 5.25 | $\geq 4.77$ | 4.0 | 1.25 |

[1]The beginning prion titer (log10) per gram of 100% prion infected brain determined by mouse bioassay for each prion strain.

not correspond to less sCJD infectivity on the wire. We are not certain why this is, but postulate that sCJD prions bind the wire with higher affinity, as different prion strains have been documented to bind metals differently [18]. Unfortunately, we did not reach an endpoint in our control wire bioassay (Table 7). But even if an endpoint had been reached, the non-linear survival times with the sCJD control wires made estimation of reductions in titers unreliable for this experiment.

While identification of the key chemical responsible for inactivation of prions was not a primary goal of our project, we believe that data from our current study combined with previous work identifies the likely anti-prion phenol. Only two phenol derivatives are present in Wex-cide 128, BP at 3.03% and OPP at 3.4%. Previous work showed that a phenolic mixture that contained OPP at 7.7%, but no BP, had no reduction in prion infectivity [4]. By deduction, the phenolic component of Wex-cide 128 that provides the most anti-prion properties is likely BP. The third phenolic derivative in eLpH was para-tertiary amylphenol (PTAP). In 2020 the U.S. Environmental Protection Agency (EPA) mandated a full phase-out of PTAP in EPA registered pesticides.

The discovery that neither eLpH or Wex-cide 128 were very effective against sCJD brain homogenates was a concern, but not entirely surprising. Previous studies have shown that different prion strains can differ in resistance to inactivation [5–10, 19]. Of particular interest were the data showing that sCJD was 10,000–100,000 times more resistant to acidic SDS inactivation compared to hamster scrapie [5, 8]. The ability of sCJD to resist acidic SDS treatment and phenolic chemicals demonstrated that sCJD may also be more difficult to inactivate using other currently approved methods.

As a biosafety precaution we reviewed the literature specific to chemical inactivation of sCJD. Several manuscripts from years ago reported concentrated bleach decreased CJD prion infectivity by 3–4 logs [20–22]. Unfortunately, the CJD tested in these studies was not directly derived from human brain, but was instead obtained from CJD that had been adapted to either guinea pigs [20, 21] or mice [20, 22]. Guinea pigs and mice both have very different PrPC amino acid sequences compared to humans. This was of great concern as we now understand that passaging a prion strain into a novel host does not guarantee the prion strain, protein folding, or susceptibility to inactivation will remain consistent with the original strain. This has been clearly demonstrated with BSE prions, where bovine BSE was shown to be 1,000 times more resistant to SDS inactivation than BSE adapted to B6 mice [5]. Unfortunately, it appears that many of the existing recommendations for inactivation of human prions were based on human prions passed through rodent models and those models may not be an accurate prediction for human prion strains [10]. Fortunately, Belondrade et. al has recently tested many chemicals against variant CJD [9, 19] and Mori et. al has shown good inactivation of sCJD prion seeding activity from steel wires using 1 M NaOH [17]. Additional studies should be performed with other prion strains to confirm that proposed or recommended decontamination methods are adequate for the targeted strain.

## Materials and methods

### Ethics statement

All mice were housed at the Rocky Mountain Laboratory (RML) in an AAALAC accredited facility in compliance with guidelines provided by the Guide for the Care and Use of Laboratory Animals (Institute for Laboratory Animal Research Council). Experimentation followed RML Animal Care and Use Committee approved protocol #2021–003-E.

## Experimental mice

Generation of tg66 transgenic mice expressing human PrP were described previously [23]. Tg66 mice were originally made by Richard Rubenstein and provided to RML by Robert Rohwer. Tg66 mice are on an FVB/N genetic background and are homozygous for a transgene that encodes human prion protein M129. Tg66 mice overexpress human PrP at 8–16-fold levels higher than normal physiologic levels and have been shown to be susceptible to human variant CJD, sCJD and mouse-adapted 22L scrapie [23, 24]. Tg66 mice do not express any mouse prion protein.

Tg33 mice express mule deer prion protein at 1-2x physiologic levels and their construction has been described previously [25]. Tg33 mice also do not express any mouse prion protein but are highly susceptible to CWD-prions [23, 25].

Tga20 mice [26] were originally obtained from the European Mouse Mutant Archive and have been partially backcrossed in-house to a C57BL/10 background. Tga20 homozygous mice over-express mouse prion protein by 5-10-fold and were used for the 22L scrapie experiments as they are highly susceptible to mouse-adapted prion agents.

Tg7 mice overexpress hamster PrPC (5-fold compared to Syrian hamster) in the absence of mouse PrPC and their construction has been described previously [27, 28]. Tg7 mice are highly susceptible to infection with the hamster 263K prion agent and develop clinical prion disease within 50days following intracranial (ic) inoculation of high titer 263K [29].

## Decontamination and bioassay of prion-infected brain homogenates using either eLpH or Wex-cide 128

Ten percent (w/v) brain homogenates (BH) were made from 263K scrapie-infected hamsters, 22L scrapie-infected C57BL/10 mice, a pool of CWD-infected white-tailed deer (WTD-1) [23], or sCJD-infected tg66 mouse brains using a mini-bead beater and 1.0 mm glass beads (Biospec products). Following homogenization, tissues were aliquoted and frozen for future use. For the decontamination studies, we used preparations of 2% eLpH (vol/vol), 4% Wex-cide 128 (vol/vol) or 40% Wex-cide 128 (vol/vol). Compared to 2% eLpH, we used 4% Wex-cide 128 to achieve approximately the same level of BP in each product (Table 1). For the decontamination, brain homogenates were thawed, vortexed and then 10 μl of each 10% brain homogenate was mixed with 90 μl of each phenolic disinfectant or saline. Total treatment time was 30 minutes, with two brief vortexes performed at 10 and 20 minutes. The resulting brain homogenate concentration during this decontamination was 1% ($10^{-2}$ dilution). Following the disinfectant treatments, the brain homogenates were further diluted in serial 10-fold increments into PBS for inoculation into mice. Due to the acute toxicity of residual Wex-cide 128 in the 40% concentration group, we could not test the $10^{-3}$ brain homogenate solution. Each dilution was inoculated intracerebrally into groups of 4–8 susceptible recipient mice. The dilutions tested for each prion strain and recipient mouse combination can be found in Tables 2–7. For the 263K experiments, tg7 mice were inoculated, 22L prions were inoculated into tga20 mice, CWD was inoculated into tg33 mice and sCJD was inoculated into tg66 mice. For the inoculation, mice were anesthetized with isoflurane and inoculated in the left-brain hemisphere with 30 μl of disinfectant-treated or saline-treated brain homogenate dilutions.

## Decontamination and bioassay of 263K or sCJD coated steel wires

Sterile stainless steel suture wires (Havel, size 000), cut into 3–4 mm lengths, were immersed in either 263K or sCJD 10% brain homogenates for one hour with gentle agitation. Following immersion, the prion infected brain homogenates were removed using a pipette and the wires

were washed briefly in an excess of sterile water. The water was drawn off and the wires were allowed to air dry in a sterile petri dish. To decontaminate the wires, wires were submerged in disinfectants (4% Wex-cide 128 or 2% eLpH) for either 2 or 30 minutes. Saline was used as a mock disinfectant. To create standard curves for the levels of prion infectivity able to bind to the steel wires, wires were exposed to ten-fold dilutions of 263K brain homogenate ($10^{-1}$–$10^{-7}$) or sCJD brain homogenate ($10^{-1}$–$10^{-5}$). Wires for each experimental group were put into 3–8 recipient mice as shown in Tables 3 and 7. 263K treated wires were implanted into tg7 mice while sCJD treated wires were implanted into tg66 mice. Wire implantation and pain management was performed as previously described [11].

## Clinical observations

All experimental mice were observed once daily by animal care staff and 3–5 times per week by prion investigators for assessment of overall health and observation for neurologic signs consistent with prion infection. Mice were euthanized when they developed clinical signs consistent with prion infection or unrelated conditions necessitating a humane endpoint (e.g. cancer, dermatitis, respiratory difficulty, chronic ocular lesions). The experiments were ended at ~300 days post-inoculation (dpi) for tg7 mice and tga20 mice and ~650 dpi for tg33 and tg66 mice. At these extended incubation periods it becomes very unlikely to see many additional mice succumb to prion infection within the described models.

## Confirmation of prion infection

In the mouse bioassays, experimental mice that were part of untreated or saline control groups that showed clear signs of clinical prion disease at the expected incubation times, were recorded as prion positive and brains from only a subset of these mice were collected. Brains from nearly all mice that were part of the disinfected groups, or brains from mice in the control groups at dilutions near the endpoint of a titration were screened for evidence of prion disease to confirm infection status (S1 Table). Screened mice that showed clinical signs of prion disease and had evidence of prion infection were scored as positive in the Tables 2–7. Different screening methods were used for the four different mouse models (below paragraphs). Mice that did not have clinical signs consistent with end-stage prion disease but did have evidence for prion disease based on a prion screening test were not included as positive mice in the bioassay tables. This subclinical situation was very rare, and only occurred with three individual mice that were part of shelf-life experiments (S1 Table).

Brains from tg7 and tga20 mice were screened by immunoblot for the presence of protease resistant PrPSc using anti-PrP antibody D13 as previously described [30, 31]. Briefly, brain homogenates were digested with 50 μg/mL proteinase K in weak detergents for 45–60 minutes. Digested samples were run on Novex Wedgewell 12% Tris-Glycine (Invitrogen) gels and the probed with D13 at a 1:50 dilution, followed by anti-human secondary antibody at 1:10,000 then developed with ECL.

Tg33 mouse brains were screened by RT-QuIC assay (methods below) for the presence of prion seeding activity. Most tg66 mouse brains were screened for neuropathology (H&E sections) and IHC demonstration of prion deposition consistent with prion disease using antiprion antibody 3F4 as described [32]. A small subset of tg66 mice that did not have formalin fixed tissue available were screened by RT-QuIC assay (methods below) for prion seeding activity rather than neuropathology.

## Calculation of prion infectivity titers

Infectivity titers were calculated for each experimental group using the Spearman-Kärber formula [33]. In experiments where no mice succumbed to prion disease at the most concentrated dilution tested, we assumed a worst-case scenario and estimated that 100% of mice would have developed disease when inoculated with a 10-fold more concentrated dose. Using this estimated data allowed the formula to be used in situations where no data was obtained above the limit of detection. In the tables, calculations resulting from data using an estimated outcome are shown with a $\leq$ sign. Prion infectivity titers for the brain homogenate experiments are reported as the $\log_{10}LD_{50}$ per gram of brain tissue. Titers for the wire experiments performed in tg7 and tg66 mice are reported as $\log_{10}LD_{50}$ per wire.

## RT-QuIC assay

RT-QuIC reactions were performed on tg33 and tg66 brains to confirm prion infection status. The RT-QuIC reaction was performed as previously described [12] using recombinant hamster 90–231 (Ha rPrP) (accession no. KO2234) as the substrate and a running temperature of 50 degrees Celsius.

The plate reader gain was consistent for each run, as were the concentrations of thioflavin T and SDS in each reaction. The maximum fluorescence readout on our plate reader is 260,000 units. For all runs, the gain was set at 1600. Four replicate wells from the same positive control mouse brain homogenate were run on each plate. In addition, 4–12 negative control wells were run at a $10^{-3}$ brain homogenate dilution on each plate. Individual wells were considered positive if they reached a fluorescence level greater than 10% of the average fluorescent values measured for the positive control wells prior to the 30-hour time point. Compared to baseline negative control wells, an increase in fluorescence from baseline to reach levels equivalent to the 10% value of positive control samples was typically 20–40 standard deviations above baseline fluorescence levels. Individual mice were scored positive if $\geq$ 50% of the assay wells were positive.

## Supporting information

**S1 Table. Summary of prion diagnostic assay results on brain tissues from bioassay mice.**
(DOCX)

## Acknowledgments

We thank Katie Williams and James Striebel for critical review of the manuscript; Robert Rohwer and Richard Rubenstein for the tg66 transgenic mice and Jeffrey Severson for many years of animal husbandry. We thank Nick Hidell of Quip Laboratories for invaluable knowledge and expertise regarding Wex-cide 128 formulation chemistries.

## Author Contributions

**Conceptualization:** Barry Twardoski, Suzette Priola, Bruce Chesebro, Brent Race.

**Formal analysis:** Chase Baune, Bradley R. Groveman, Suzette Priola, Brent Race.

**Funding acquisition:** Bruce Chesebro.

**Investigation:** Chase Baune, Bradley R. Groveman, Tina Thomas, Brent Race.

**Resources:** Andrew G. Hughson.

**Supervision:** Brent Race.

**Writing – original draft:** Brent Race.

**Writing – review & editing:** Chase Baune, Bradley R. Groveman, Barry Twardoski, Suzette Priola.

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
