## [Decision Letter · Decision Letter 0]

28 Jul 2023

PONE-D-23-20329Efficacy of Wex-cide 128 disinfectant against multiple prion strainsPLOS ONE

Dear Dr. Race,

Thank you for submitting your manuscript to PLOS ONE. After careful consideration, we feel that it has merit but does not fully meet PLOS ONE’s publication criteria as it currently stands. Therefore, we invite you to submit a revised version of the manuscript that addresses the points raised during the review process.

We look forward to receiving your revised manuscript.

Kind regards,

Gianluigi Zanusso

Academic Editor

PLOS ONE

Journal Requirements:

“This research was supported by the Intramural Research Program of the NIH, National Institute of Allergy and Infectious Diseases.”

Additional Editor Comments:

Authors show that Wex-cide 128 is an efficacious prion disinfectant and not caustic for the equipment. Based on the present findings, might Wex-cide 128 be suitable for neurosurgical instruments prion contamination ? or folllowing decontamination the instruments might be not reusable?

Reviewers' comments:

Reviewer's Responses to Questions

**Comments to the Author**

1. Is the manuscript technically sound, and do the data support the conclusions?

Reviewer #1: Yes

Reviewer #2: Yes

2. Has the statistical analysis been performed appropriately and rigorously? 

Reviewer #1: Yes

Reviewer #2: Yes

3. Have the authors made all data underlying the findings in their manuscript fully available?

Reviewer #1: Yes

Reviewer #2: Yes

4. Is the manuscript presented in an intelligible fashion and written in standard English?

Reviewer #1: Yes

Reviewer #2: Yes

5. Review Comments to the Author

Reviewer #1: This is an interesting and useful study that demonstrate the efficay of the chemical disinfectant Wex-cide 128 on some strains of natural and experimental prion diseases. Several conditions of treatment have been compared with reference to the discontinued prion-inactivation product Environ LpH. Authors know very well the subject and their data are clearly presented, analysed and discussed. The only point of weakness may reside in the relatively small number of animals dedicated to some steel wire assays (tables 3 and 7) with only 4 or 6 inoculated mice. In these assays, no other close dilutions have been tested. According to this reviewer's experience, a significant number of animals is particularly important when dealing with limited infectivity where the absence, rather than the presence, of prions has to be demonstrated. Authors may improve the manuscript by adding a comment on that (or, if they prefer, by evaluating the statistical power of their assay).

Some notes (mostly typos) are in the attached pdf.

Reviewer #2: In this manuscript the authors examine the efficacy of Wexide-128 to inactivate several strains of rodent-adapted prions and to a clinical isolate of sporadic CJD. The experiments are well-designed and properly controlled. The manuscript is clearly written and the conclusions are supported by the data. This is a significant study in that it identifies another means of inactivating prions and it provides further evidence that strain specific differences in inactivation occur. This reviewer only has minor suggestions.

In table 4, the saline treated material did not reach an attach rate below 50%, it is unclear how an LD50 was calculated.

6. PLOS authors have the option to publish the peer review history of their article (what does this mean?). If published, this will include your full peer review and any attached files.

Reviewer #1: **Yes: **Franco Cardone

Reviewer #2: No

---

## [Author Response · Author response to Decision Letter 0]

1 Aug 2023

Response to Reviewers

The authors would like to thank the reviewers for the helpful comments and recommendations. We have revised the manuscript based on their input. Responses to individual concerns are provided below. Changes to the manuscript are best viewed using the marked-up copy with track changes. 

Additional Editor Comments:

Authors show that Wex-cide 128 is an efficacious prion disinfectant and not caustic for the equipment. Based on the present findings, might Wex-cide 128 be suitable for neurosurgical instruments prion contamination? or following decontamination the instruments might be not reusable?

Wex-cide 128 was very efficacious against eliminating animal TSE infectivity, but much less effective at removing sCJD prions from brain homogenates. The inactivation of sCJD infectivity from steel wires was encouraging, but still not 100% with a 2-minute treatment, so based on the homogenate data and wire data, we would not recommend Wex-cide 128 for treatment of instruments used for human procedures. The apparent resistance of sCJD to Wex-cide is unfortunate. 

5. Review Comments to the Author

Reviewer #1: This is an interesting and useful study that demonstrate the efficay of the chemical disinfectant Wex-cide 128 on some strains of natural and experimental prion diseases. Several conditions of treatment have been compared with reference to the discontinued prion-inactivation product Environ LpH. Authors know very well the subject and their data are clearly presented, analysed and discussed. 

Thank you for the very positive and supportive comments.

The only point of weakness may reside in the relatively small number of animals dedicated to some steel wire assays (tables 3 and 7) with only 4 or 6 inoculated mice. In these assays, no other close dilutions have been tested. According to this reviewer's experience, a significant number of animals is particularly important when dealing with limited infectivity where the absence, rather than the presence, of prions has to be demonstrated. Authors may improve the manuscript by adding a comment on that (or, if they prefer, by evaluating the statistical power of their assay).

Some notes (mostly typos) are in the attached pdf.

We agree with R1, small group sizes can be problematic with limited infectivity. Fortunately, in our wire experiments our untreated controls were highly infectious, providing support to the conclusion that the absence of infectivity in decontaminated wires was evidence for efficacy. We have added text to lines 325-329 to point out the weakness of performing assays on only a single dilution and small group sizes. 

Typos indicated in the PDF have been corrected in the revised manuscript. Other responses to the PDF comments in brief: Table 2, N=7 was corrected to N=8 for the inconsistency noted. R1 was also correct in catching our PK concentration typo in line 449, mg should be °g. We also did confirm the range of group sizes reported in the methods was accurate for the wire experiments was 3-8 mice (comment in line 420 of PDF). Table 3 has a maximum group of 8, table 7 has a group as small as 3. 

Reviewer #2: In this manuscript the authors examine the efficacy of Wexide-128 to inactivate several strains of rodent-adapted prions and to a clinical isolate of sporadic CJD. The experiments are well-designed and properly controlled. The manuscript is clearly written and the conclusions are supported by the data. This is a significant study in that it identifies another means of inactivating prions and it provides further evidence that strain specific differences in inactivation occur. This reviewer only has minor suggestions.

In table 4, the saline treated material did not reach an attach rate below 50%, it is unclear how an LD50 was calculated.

Thank you for your supportive comments. Yes, it is unfortunate that our bioassay of the saline treated CWD did not reach an end-point. In this case we assumed the next ten-fold dilution (10-8) would have been 0/4 positive mice and calculated the titer based on the data shown through 10-7. A footnote (b) has been extended to describe this caveat (lines 210-211) and a ≥ sign placed in front of the reported titer in table 4, as we may have made an underestimate.

---

## [Editor Report · Decision Letter 1]

4 Aug 2023

Efficacy of Wex-cide 128 disinfectant against multiple prion strains

PONE-D-23-20329R1

Dear Dr. Race,

We’re pleased to inform you that your manuscript has been judged scientifically suitable for publication and will be formally accepted for publication once it meets all outstanding technical requirements.

Kind regards,

Gianluigi Zanusso

Academic Editor

PLOS ONE

---

## [Editor Report · Acceptance letter]

15 Aug 2023

PONE-D-23-20329R1 

Efficacy of Wex-cide 128 disinfectant against multiple prion strains 

Dear Dr. Race:

I'm pleased to inform you that your manuscript has been deemed suitable for publication in PLOS ONE. Congratulations! Your manuscript is now with our production department. 

Kind regards, 

on behalf of

Dr. Gianluigi Zanusso 

Academic Editor

PLOS ONE